# Preventive Role of L-Carnitine and Balanced Diet in Alzheimer’s Disease

**DOI:** 10.3390/nu12071987

**Published:** 2020-07-03

**Authors:** Alina Kepka, Agnieszka Ochocinska, Małgorzata Borzym-Kluczyk, Ewa Skorupa, Beata Stasiewicz-Jarocka, Sylwia Chojnowska, Napoleon Waszkiewicz

**Affiliations:** 1Department of Biochemistry, Radioimmunology and Experimental Medicine, The Children’s Memorial Health Institute, 04-730 Warsaw, Poland; e.skorupa@ipczd.pl; 2Department of Pharmaceutical Biochemistry, Medical University of Bialystok, 15-089 Bialystok, Poland; malgorzata.borzym-kluczyk@umb.edu.pl; 3Department of Medical Genetics, Medical University of Bialystok, 15-089 Bialystok, Poland; beata.stasiewicz.jarocka@gmail.com; 4Faculty of Health Sciences, Lomza State University of Applied Sciences, 18-400 Lomza, Poland; sylwiacho3@gmail.com; 5Department of Psychiatry, Medical University of Bialystok, 15-089 Bialystok, Poland; napwas@wp.pl

**Keywords:** Alzheimer’s disease, L-carnitine, carnitine supplementation, Mediterranean diet, MIND diet

## Abstract

The prevention or alleviation of neurodegenerative diseases, including Alzheimer’s disease (AD), is a challenge for contemporary health services. The aim of this study was to review the literature on the prevention or alleviation of AD by introducing an appropriate carnitine-rich diet, dietary carnitine supplements and the MIND (Mediterranean-DASH Intervention for Neurodegenerative Delay) diet, which contains elements of the Mediterranean diet and the Dietary Approaches to Stop Hypertension (DASH) diet. L-carnitine (LC) plays a crucial role in the energetic metabolism of the cell. A properly balanced diet contains a substantial amount of LC as well as essential amino acids and microelements taking part in endogenous carnitine synthesis. In healthy people, carnitine biosynthesis is sufficient to prevent the symptoms of carnitine deficiency. In persons with dysfunction of mitochondria, e.g., with AD connected with extensive degeneration of the brain structures, there are often serious disturbances in the functioning of the whole organism. The Mediterranean diet is characterized by a high consumption of fruits and vegetables, cereals, nuts, olive oil, and seeds as the major source of fats, moderate consumption of fish and poultry, low to moderate consumption of dairy products and alcohol, and low intake of red and processed meat. The introduction of foodstuffs rich in carnitine and the MIND diet or carnitine supplementation of the AD patients may improve their functioning in everyday life.

## 1. Introduction

Recent progress in the medical sciences have significantly prolonged human life and the incidence of old age diseases. Old age diseases are a rapidly growing cause of disability and/or death [1]. There was proof that some diet supplements may have a positive therapeutic effect on patients with neurodegenerative diseases, particularly in AD [2]. Recently, the attention of physicians as well as their patients is increasingly directed to dietary supplements, which have become a more and more attractive option in neurodegenerative disease treatment [3].

Neurodegenerative diseases, including AD, are characterized by a loss of neurons and synapses of the brain cortex and some subcortical regions, resulting in the atrophy and degeneration of the involved regions in the temporal and parietal lobes as well as parts of the frontal lobe and part of the callosal gyrus [4,5]. The extracellular deposition of amyloid-β, the intracellular deposition of protein tau (τ) and microglia activation are causes of AD’s pathological symptoms [6]. Recently, it has been reported that an essential role in the pathogenesis of AD may be also played by other proteins such as α-synuclein and the protein TDP-43 [7]. It has been confirmed that there are genetic predispositions to the onset of AD. People with the defective e4 variant of the apolipoprotein E gene, which is involved in amyloid-β degradation, may already have symptoms of AD in the early period of life [8,9]. The accumulation of β-amyloid plates in neurons decreases the efficiency of electron transport in the respiratory chain, leading to a decrease in ATP production, the induction of oxidative stress and the disturbance of Ca^++^ homeostasis. On the other hand, the accumulation of the protein τ inside neurons blocks the intracellular transport of proteins, nutrients and neurotrophins. The primary symptoms of AD present as disturbances of recent memory, the concentration of attention and orientation. The leading symptoms of AD are aphasia (speech disturbance), apraxia (inability of movement despite non-handicapped motoric functions), agnosia (lack of sensory ability to recognize objects despite correct sensory function), and disturbances in sleep and wakefulness. The deepening symptoms in AD patients, connected with disturbances in simple and complex everyday life activities, make independent existence impossible. Ophthalmic symptoms—e.g., the impairment of chromatic vision, scotoma, the reduction of contrast sensitivity, disturbances in the mobility of the eyeballs, and retina degeneration—also occur in AD. The deposition of β-amyloid concrements in the eyes of AD patients were demonstrated through eye ground imaging [10]. 

Acetylcholine (Ach) plays a crucial role in the cognitive functions of the brain [5]. A decrease in the activity of the mechanisms responsible for the biosynthesis and degradation of Ach, such as choline acetyltransferase, acetylcholine esterase, transport system with high affinity to choline (HACU) and follicular acetylcholine transporter (VAChT), was found in patients with AD [5]. The impairment of the transmission of nervous signals caused by a decrease in the density of muscarinic and nicotinic receptors, a decrease in the intraneural concentration of Ach and Ach accumulation in synaptic vesicles was demonstrated in AD [7,11,12]. Numerous articles document disturbances in the biosynthesis and metabolism of carnitine in AD patients [13,14,15]. Low concentrations of free carnitine, acetyl-L-carnitine (ALC) and other acylcarnitines were found in plasma and tissues in many studies on AD [13,16]. A progressive decrease in ALC and other acyl-carnitine serum levels in healthy subjects (HS) through to subjective memory complaints (SMC) and mild cognitive impairment (MCI) up to Alzheimer’s disease (AD) were reported. ALC significantly decreased on average by 21% in SMC, 27% in MCI and 36% in AD as compared to in HS [16]. The deficit of ALC suggests a perturbed transport of fatty acids into the mitochondria for beta-oxidation, as well as suppressed energy metabolism. The results of transcriptomic studies showing a significant decrease in the activity of the carnitine shuttle in AD patients are consistent with the above-described hypothesis. The deficit of the carnitine shuttle might contribute to the mitochondrial dysfunctions supposed to be responsible for many neurodegenerative diseases including AD. The decreased serum levels of some acyl-carnitines found in MCI subjects might indirectly signal an impending progression of dementia and might be used as biomarkers of phenotype conversion from MCI to AD [16]. The results described above suggest that serum ALC and other acyl-L-carnitine levels decrease along a continuum from HS to SMC and MCI subjects, up to patients with AD [13,14,16]. It was demonstrated that ALC facilitates cholinergic neurotransmission directly or by providing an acetyl group that may be used for acetylcholine synthesis [17,18]. It was reported that ALC, by the stimulation of the synthesis of nerve growth factor receptors in the hippocampus and basal forebrain, prevents the loss of muscarinic receptors as well as nerve growth factor (NGF) and directly or indirectly modulates N-methyl-D-aspartate receptor (NMDA). The excessive stimulation of NMDA receptors by glutamic acid may induce the uncontrolled influx of Ca^++^ into cells, causing damage to and the death of neurons [19]. In addition, it was proven that carnitine significantly increases dopamine levels in the cortex, hippocampus and striatum of the rat brain [20]. In animal experiments, it was found that ALC supplementation prevents the hyperphosphorylation of the protein τ induced by homocysteine (a new marker of AD) and inhibits the phosphorylation of β-amyloid [21]. It has been shown that ALC and L-carnitine reduce apoptosis through the mitochondrial pathway [18]. Suchy et al. [22] reported that carnitine supplementation reduces damage to the murine brain caused by free radicals and improves cognitive performance. A beneficial effect of ALC on cognition and behavior in aging and AD subjects was reported [23]. The possible mechanisms of ALC action in AD may also involve facilitating the rebuilding of cell membranes, as well as improving synaptic function, enhancing cholinergic activity, restoring the brain energy supply, protecting against toxins, and exerting neurotrophic effects via stimulating NGF and the acetylation of proteins [24].

The introduction of an appropriate balanced diet, besides carnitine supplementation (especially ALC), is an important factor that can delay or prevent the onset of AD. Many studies have suggested that the diet MIND (Mediterranean-DASH Intervention for Neurodegenerative Delay)—composed of the DASH (Dietary Approaches to Stop Hypertension) diet and the Mediterranean diet, considered one of the healthiest diets on the earth—appears to be the most effective in preventing neurodegeneration, especially Alzheimer’s disease. [25]. Adherence to the Mediterranean diet may not only reduce the risk of AD but also diminish pre-dementia syndromes and their progression to overt dementia. Based on the current evidence, there are no definitive dietary recommendations for the prevention of AD. However, the following dietary advice for lowering the risk of AD and inhibiting cognitive decline, as well as decreasing all-cause mortality in AD patients, is suggested: a high level of consumption of fats from fish, vegetable oils, non-starchy vegetables, and low glycemic index fruit; a diet low in foods with added sugars; and a moderate wine or beer intake should be encouraged [26].

## 2. AD Epidemiology

Contemporary society aging is connected with the occurrence of neurodegenerative disorders leading to dementia, creating serious problems in complex medical and social care. It was estimated that in 2006, 7.3 million (mln) Europeans (12.5 per 1000 residents) in 27 member states, between 30 and 99 years of age, suffered from different kinds of dementia. Among Europeans, dementia affected women (4.9 mln) more frequently, than men (2.4 mln). Together with an increase in the average life span, especially in developed countries, the frequency of dementia has dramatically increased. The number of people with dementia worldwide is expected to double every 20 years, reaching almost 75 million by 2030 [27]. Most recent regional estimates of the age-standardized prevalence of dementia in people aged 60 and over fall between 5.6 and 7.6%. Although dementia was traditionally viewed as more prevalent in developed countries, these estimates range from 4.6% in Central Europe to 8.7% in North Africa and the Middle East [28]. Currently, there are over 70 million people in the world’s oldest group, over 80 years. It is estimated that by 2050, the population over 80 will increase several times. According to data from the World Health Organization (WHO), in 2030, there will be 65 million; in 2040, there will be 80 million; and in 2050, there will be 115 million dementia patients [29,30]. The seven places with the largest numbers of people with dementia in 2001 were as follows: China (5.0 mln), the European Union (5.0 mln), the USA (2.9 mln), India (1.5 mln), Japan (1.1 mln), Russia (1.1 mln) and Indonesia (1.0 mln) [30]. Currently, in the developed countries, the population of people over 65 constitutes 14% (about 170 mln), while in the developing countries, it is about 5% (about 248 mln). According to the expected demographic changes, it is estimated that in 2030, people over 65 will constitute 23% (275 mln) in developed countries and 10% (680 mln people) in developing countries. It is estimated that AD may affect about 6% of the people of the world over 65 and 40% of people over 85 [31]. It is estimated that by 2050, there will be in the world two billion people aged 60 years or over, of which 131 million will be affected by dementia, but depression will be the second cause of disability worldwide by 2020. Preventing or delaying the onset of dementia and depression should therefore be a world public health priority [29,30,32]. Today, in the world, almost 44 million people suffer from AD, and each year, 4.5 million new cases of dementia are diagnosed. In the world, the risk of dementia doubles every 5 years, with AD responsible for 2/3 of all cases of dementia. In the USA, AD is placed in third position after neoplasms and cardiovascular diseases concerning social security costs [31]. 

For Poland’s demographics, prognoses as Poland’s population gets old are also not optimistic, and it has been estimated that in the next 25 years, the average age of men will increase by 7.2 years and that of women, by 4.5 years. It should be mentioned that in 2010, the expected length of life in Poland averaged 75.9 years (men, 71.9 years; women, 80.1 years). According to the official Polish data from 1999, population studies were conducted in two centers, among the population of the Warsaw district of Mokotów and in Świebodzin (rural and small town communities). The epidemiological data from Świebodzin revealed that the prevalence of AD in the population over 65 was 3.5% and that of vascular dementia, 3.6%. In the second study (Mokotów, Warsaw district, Poland), dementia was found in the population from 65 to 84 years of age to be at the level of 5.7% (including AD at 2.3% and vascular dementia at 2.7%) [33]. Population data published in 2007 by Bdzan et al. [34] regarding the studied rural populations (Pruszcz Gdański, Trąbki Wielkie and Pszczółki, Poland) showed that the prevalence of dementia was estimated at 6.7%. The prevalence of dementia in total rural populations was 3.0% for men and 8.8% for women. The prevalence of Alzheimer’s disease was 1.1% for men and 4.0% for women, and that of vascular dementia, 1.9% and 3.5%, respectively. Bdzan et al.’s [34] study presents data showing that rural populations had a higher incidence of dementia disorders than that previously reported in other Polish regions and that women had a higher dementia risk than men. It should be stressed that in 2005–2010, the annual increase in the number of dementia patients remained levels not exceeding 2%, but in 2010–2015, the annual increase in the number of dementia patients reached 3–4%. The Polish National Statistics Agency predicts an increase in the number of people reaching retirement age to 23.8% of the Polish population (4.8 million) by 2030 [35]. According to the Polish National Statistics Agency report of 2016, presently in Poland, people above 65 constitute 14.7% of the population, and increases to 24.5% by 2035 and to above 30% by 2050 are predicted. In Poland, AD presently affects more than 300,000 people, mostly elderly people, and it is estimated that the number of AD subjects will triple by 2050, reaching almost a million [36]. 

The above figures show the importance of the problem facing health services not only in Poland but also over the whole world [37]. It should be taken into consideration that the further aging of the Polish population will increase the numbers of people that take advantage of medical and social care, are disabled and suffer from non-infectious chronic diseases. In the existing situation, the Polish system of health care—similarly to the systems in other countries—must take up the challenge of creating an efficient system of care for people suffering from dementia.

## 3. Prevention of Alzheimer’s Disease—Mediterranean and MIND Diets

In contemporary medicine, the prevention of neurodegenerative diseases is one of the main objectives for many branches of health services. The appropriate types and quality of food may be important factors for preventing and supporting the treatment of AD. The early prevention of AD starts by reducing risk factors: reducing the smoking of cigarettes, preventing hypertension, insuring optimal concentrations of homocysteine, preventing type 2 diabetes, combating insulin resistance and obesity, limiting stress, avoiding toxins, and mental and physical training are important components of AD prevention. Appropriate nutrition is an essential and modifiable factor that plays a key role in preventing and/or delaying the onset of dementia, including AD. It was reported that a reduction in amount of fried meat and an increase in the amounts of other foods (such as fish, cheese, vegetables and vegetable oil) in the diet significantly reduced the incidence of AD [38]. Diets rich in advanced glycation end products (AGEs), which arise during long-lasting food thermal processing (heating, frying and irradiation), significantly accelerate the development of AD [39]. High concentrations of AGEs are contained in (a) products containing sugar (candy, cookies, chocolate biscuits, cakes, fizzy drinks, pastries and sauces); (b) processed meats such as sausages and conserved and preserved meat; (c) processed dairy products; (d) food containing trans fats such as margarine and cream; and (e) highly fried products such as fried potatoes, crisped cakes etc. The lowest amounts of AGEs are found in fresh fruits, vegetables, seafood, products with short thermal processing, and raw and non-processed food. A correctly composed diet rich in polyunsaturated (from the families of omega-3 and omega-6) and monounsaturated fatty acids as well as antioxidative vitamins (E, C and β-carotene) reduces the risk of AD development. It was reported that a decrease in the impairment of cognitive function and dementia risk were connected with a decrease in the consumption of milk and dairy products. The consumption of full-fat dairy products may be connected with worsening cognitive function in older people. It was reported that moderate alcohol consumption may be connected with a decrease in the risk of dementia due to AD [40,41]; however, recently, the Lancet published a statement concerning alcohol, that there is “no safe limit, even one drink a day, increases risk of chronic non infectious diseases” [42].

A large study demonstrated the association of consuming a Mediterranean diet with a decrease in the incidence of AD, which creates hope for using the Mediterranean diet as a modifiable risk factor in protection against AD [43,44]. A Mediterranean diet rich in vegetables with low starch contents, fruit with low glycemic indices, cereal products, legumes, plant oils (olive, colza, linen and sunflower) and fish (especially sea fish: halibut, herring, mackerel and sardines) and a diet containing moderate amounts of meat and dairy products positively affect health conditions and may decrease the risk of the development of many diseases including neurodegenerative diseases [25,45].

It was reported that people adhering to a Mediterranean diet have a 28% decreased risk of cognitive disturbances and 48% decreased risk of AD in comparison to people who do not consume a Mediterranean diet [46]. It seems that a complex nutritional strategy initiated in the early stages of cognitive impairment is the most pragmatic approach for controlling the progress of AD [47]. Presently, for the elderly and people at risk of AD, nutrition based on the MIND (Mediterranean-DASH Intervention for Neurodegenerative Delay) diet—composed of DASH (Dietary Approaches to Stop Hypertension) diet and the Mediterranean diet (MD), considered as the healthiest diet on earth, constituting a careful nutritional program—is recommended [48]. In our opinion, the MIND diet should be a very important element of many disease prevention strategies, including those for dementia and AD (Table 1).

Mediterranean and Asian diets are currently considered the healthiest. Good eating habits are effective in combating the risk of age-related diseases, especially cardiovascular and neurodegenerative diseases. Foods and drinks of plant origin such as green tea, extra-virgin olive oil, red wine, spices, berries and aromatic herbs have beneficial effects in the prevention of amyloid diseases, e.g., Alzheimer’s disease, Parkinson’s disease and prion diseases. Mediterranean and Asian diets are becoming more and more attractive for the prevention and treatment of neurodegenerative diseases, as it has been proven that they can inhibit the production of amyloidogenic peptides, increase the activity of antioxidant enzymes, activate autophagy and reduce inflammation [49] (Table 1). Currently, some foods or food groups traditionally considered harmful such as eggs and red meat have been partially rehabilitated, but there is still a negative correlation of cognitive functions with saturated fatty acids. Protective effects against cognitive decline of elevated fish consumption and a high intake of monounsaturated fatty acids and polyunsaturated fatty acids (PUFA), particularly n-3 PUFA, was confirmed [40]. Cheese and yoghurt should be moderately consumed, while meat should be rarely consumed. Wine or beer (mostly non-alcoholic beer) during main meals are also one of the components of the Mediterranean diet. Beer consumption, and its content of bioavailable silicon, reduces the accumulation of aluminum in the body and brain tissue and lipid peroxidation, and protects the brain against neurotoxic effects by regulating antioxidant enzymes [26].

## 4. Physiological Properties of L-Carnitine

L-carnitine (2-hydroxy-4-trimethylammonium butyrate) (LC) plays many important roles in the intracellular functions of the body. The most important role is played by obtaining cellular energy from fatty acids (FAs) in the mitochondrial matrix and maintaining mitochondria coenzyme A (CoA) homeostasis during mitochondrial FA oxidation [50]. Maintaining the mitochondrial homeostasis of CoA is an extremely important function of carnitine in supporting optimal cellular CoA and acyl-CoA concentrations. CoA is necessary for the activation and oxidation of the FAs from adipose tissue in the mitochondria for ATP synthesis. FA oxidation reduces glucose oxidation in the tissues where glucose is not an essential fuel, and amino acid catabolism for gluconeogenesis and energy production. FAs are a very efficient source of human energy, as the yield of energy from the total oxidation of FAs is 37.7 kJ/g as compared to 16.7 kJ/g from protein or carbohydrates. Another role of carnitine in human metabolism is participation in the processes of the detoxication of toxic exogenous compounds (e.g., some xenobiotics, including ampicillin, valproic acid and salicylic acid), which are excreted by the kidneys in combination with carnitine [51]. The next important role of carnitine is its contribution to the catabolism of branched-chain ketoacids derived from branched-chain amino acids (valine, leucine and isoleucine). L-carnitine also inhibits free radical production and demonstrates antioxidant action [51]. 

The richest source of L-carnitine is red meat consumed by adults, and milk consumed by infants and children [52], whereas plants contain only trace of carnitine [53] (Table 2). The standard human diet covers about 3/4 of the requirements for L-carnitine, and the remaining 1/4 is synthesized in the human body from lysine and methionine with the participation of ascorbic acid, niacin, piridoxin and Fe^+2^ [54] in the liver, kidneys, brain [50] and placenta [55]. Nutritional carnitine is actively (sodium dependent) and passively transported from the intestinal contents into enterocytes, with 54–86% bioavailability, depending on the amount of carnitine in a meal. The bioavailability of carnitine from dietary supplements (0.5–6.0 g) is much lower than that from the diet and reaches only 14–18%. It should be mentioned that organisms maintain carnitine homeostasis, and along with an increase in carnitine consumption, there is decrease in carnitine absorption [56]. Consumed but not absorbed carnitine is degraded, mainly by intestinal microorganisms to trimethylamine (TMA), which can reach the liver, where it is transformed into the toxic trimethylamine N-oxide (TMNO), excreted in the urine. To date, TMA and TMNO have mostly been treated as nontoxic substances, but more recently, they have been treated as potentially carcinogenic agents, because of the possibility of their transformation into N-nitrosodimethylamine (NDMA) [57]. However, no evidence of clinical implications of TMAO in the central nervous system has been documented as yet, but TMAO is present at detectable levels in the cerebro-spinal fluid (CSF). In a small tested groups of subjects, TMAO levels in the CSF were apparently unrelated to the diagnosis of neurological disorders such as AD [58]. 

In the human intestine, carnitine is acetylated to biologically active acetyl-L-carnitine (γ-trimethyl-β-acetylbutyre-betaine) (ALC). Since ALC is transferred through intestinal serous membranes better than non-acetylated L-carnitine, the intracellular acetylation of carnitine may facilitate its diffusion across the serosal membrane [56]. LC and its short chain fatty esters do not combine with plasma proteins, though blood cells contain LC, but the speed of the transport of LC between erythrocytes and the plasma is very slow [59]. In the circulation, about 75% of LC occurs in a free state, 15% occurs as ALC, and the remaining 10% occurs as esters of carnitine with other acids (e.g., propionyl-L-carnitine) [60]. In human tissues, carnitine is localized mainly in skeletal and cardiac muscles (98%), whereas in the liver, kidneys and brain resides only about 1.5% [52,61]. In the adult brain, about 80% of carnitine exists as free carnitine (FC); 10–15%, as ALC; and less than 10%, as long chain acylcarnitines [62]. The plasma of healthy people on a normal diet contains only small amounts of LC (about 0.5–1% of total body L-carnitine) [63], amounting to about 40–60 µM in plasma [59]. There are at least three pharmacokinetic compartments for LC. The first consists of extracellular fluid (including plasma), constituting the central compartment, reflecting the initial carnitine content; the second compartment is relatively small, with fast turnover, involving the liver, kidneys and other organs; the third and the greatest compartment involves the skeletal and heart muscles. The turnover times (or average times for which carnitine resides in compartments) amount to about 1, 12 and 191 h for compartments one, two and three, respectively, and the total carnitine turnover time for the whole human organism amounts to about 66 days. The velocity constant for the transport of carnitine from muscle to plasma is about 0.0005 h^−1^, which means that the half-time for carnitine transportation from muscle to plasma is about 5–6 days. Muscle carnitine content is rather stable, because the transport of carnitine to and from muscle cells is a very slow process, lasting weeks and even months [59]. Since the taking of carnitine from the muscles is a long process in carnitine deficiency, it is advisable to apply carnitine supplementation to quickly restore appropriate concentrations of carnitine in the first (extracellular fluids) and second compartments (internal organs excluding muscles) with short turnover times. It was reported that in humans, after the per os administration of carnitine at doses of 30–100 mg/kg, carnitine’s peak plasma concentrations were 27–91 μM after 3 h, and the carnitine concentrations returned to basal levels by 24 h. After the intravenous injection of single doses (0.5 g) of LC, the plasma concentrations of ALC and LC returned to their initial values by 12 h [58]. 

In the homeostasis of carnitine and its ester, the main role in the maintenance of appropriate concentrations is played by the kidneys [64]. Carnitine is not metabolized in humans but undergoes filtration in the glomeruli and is almost totally (98–99%) reabsorbed in the renal canaliculi. Sodium-dependent carnitine organic cation transporter (OCTN2) plays a key role in renal carnitine reabsorption [65]. The renal threshold for free carnitine is in the range of its physiological plasma concentration.

### 4.1. Role of L-Carnitine and Acetyl-L-Carnitine in Human Brain Metabolism

L-carnitine is actively transported to the brain through the blood–brain barrier by the organic cation transporter OCTN2 and accumulates in neural cells especially as acetylcarnitine. LC, besides its important role in the metabolism of lipids, is also a potent antioxidant (free radical scavenger) and thus protects brain tissues against oxidative damage [66]. Mitochondria, as the main source of cell energy as well as ROS and antioxidants, play a key role in the production of ATP, regulating apoptosis and detoxication, maintaining membrane potentials and the distributions of ions in appropriate compartments of the cell, etc. Maintaining mitochondrial homeostasis is crucial for the development and proper action of neurons. Dietary supplements applied for maintaining mitochondrial homeostasis include L-carnitine; coenzyme Q_10_; mitoquinone mesylate; and other mitochondrion-targeted antioxidants such as N-acetylcysteine; vitamins C, E, K_1_ and B; sodium pyruvate; and lipoic acid [67].

Even though the brain’s basic energetic substrate is glucose, LC is important in brain energetic lipid metabolism, taking part in the transport of the long chain fatty acids from the cytoplasm to mitochondria [52] and acetyl groups from mitochondria to the cytoplasm [62]. Maintaining a suitable relationship between cellular acetyl-CoA and CoA, L-carnitine ensures correct energetic cell metabolism [68]. It should be mentioned that connecting acyl and acetyl groups to LC increases LC’s hydrophobicity, which facilitates LC’s crossing of the blood–brain barrier. After the penetration of brain mitochondria with long chain fatty acids, LC reacts with free coenzyme A (CoASH)—a reaction catalyzed by carnitine palmitoyltransferase II (CPT II)—releasing LC [50] (Figure 1). 

Therefore, LC improves energetic brain homeostasis by supplying acyl groups to the mitochondria of the brain cells. ALC is an important factor expanding the brain’s sources of energy; therefore, treatment with ALC is responsible for a reduction in brain glycolytic flow and the enhancement of the utilization of alternative energy sources, such as fatty acids or ketone bodies [69], that may reduce brain glucose utilization [70]. ALC can provide an acetyl moiety that can be oxidized for energy production; used as a precursor for acetylcholine; or incorporated into glutamate, glutamine and γ-aminobutyric acid, as well as into lipids for myelination and cell growth [71]. ALC also has many other functions in the body. ALC’s ability to freely pass the brain–blood barrier could help brain fatty acid transport for oxidation in mitochondria that improves brain energy metabolism. ALC supplementation positively influences the activity of enzymes in the tricarboxylic acid cycle, the electron transport chain and amino acid metabolism [72]. ALC acts neuroprotectively by improving the energetic function of brain mitochondria, the elimination of oxidative products, the stabilization of the cell membranes and neurotrophic factor production (stimulates protein and phospholipid biosynthesis), and it exerts anti-apoptotic functions, as well as modulating the expression of genes coding proteins, and protects neural cells from excitotoxicity [73]. Additionally, ALC provides acetyl groups for the synthesis of acetylcholine [17] and acetylation of nuclear histones [74]. ALC counteracts stressogenic agents by maintaining proper plasma concentrations of β-endorphin and cortisol [75]. ALC stimulates α-secretase activity and physiological amyloid precursor protein (APP) metabolism. In particular, ALC favors the delivery of disintegrin and metalloproteinase domain-containing protein 10 (ADAM10), the most accredited α-secretase, to the post-synaptic compartment and, consequently, positively modulates ADAM10’s enzymatic activity toward APP [76]. Palmitoylcarnitine (ester of carnitine with palmitic acid) can stimulate the expression of GAP-43 (also named B-50, neuromodulin, F1, pp45), a protein involved in neural development, neuroplasticity and neurotransmission [77]. ALC increases the concentration of the brain-derived neurotrophic factor (BDNF), that is lowered in AD patients [78,79]. ALC increases astrocytes’ glutathione concentrations (an important cellular antioxidant), which are lowered with age [80]. ALC is recommended for delaying the onset and development of Alzheimer’s and Parkinson’s diseases, and alleviating the symptoms of senile depression and memory disturbances connected with age; it also improves studying and memory [23,81,82]. It was confirmed that the multidirectional positive role of ALC in the dopaminergic system [20] depended on the slowing down of the progressive deterioration of dopaminergic receptors with a simultaneous increase in the concentrations of dopamine (a neurotransmitter responsible for mood, processes of thinking, the coordination of movement and resistance to stress) in neurons [83]. It was reported that the LC supplementation of experimental animals significantly increased their levels of neurotransmitters such as noradrenaline, adrenaline and serotonin, especially in brain regions rich in cholinergic neurons, i.e., the brain cortex, hippocampus and striatum [20,84]. ALC improves dopamine metabolism, prevents degenerative changes in dopamine-producing neurons and reduces the age-dependent process of the destruction of receptors that bind dopamine [85]. Cristofano et al. [16] showed a progressive decrease in ALC and other acyl-carnitines’ serum levels in people changing from normal to AD and concluded that the decreased serum concentrations of ALC and hence its disturbed functions may predispose to AD and contribute to neurodegeneration. Clinical studies in humans demonstrated positive effects of ALC on brain function, cognition and memory that led to the suggestion that ALC may slow or reverse mild cognitive impairment and the progression of dementia in Alzheimer’s disease [58].

ALC supplementation is recommended for improving brain and nervous system action, memory, the speed of learning and memorization, the level of brain energy, psychological conditions and mood, and the effects of therapies for brain neurodegenerative disorders and peripheral neuropathies [66,81]. 

### 4.2. Recommendations for L-Carnitine Content in the Diet

LC is an essential nutritional component delivered in food produced from animals, because LC endogenic synthesis is insufficient to cover metabolic needs. Primary carnitine deficiency is rare, but secondary carnitine deficiency is more frequent, being associated with several inborn errors of metabolism and acquired medical or iatrogenic conditions, for example, in patients under valproate and zidovudine treatment. Other chronic conditions such as diabetes mellitus, heart failure and Alzheimer’s disease in connection with diseases creating increased catabolism may cause secondary carnitine deficiency [86].

Presently, there are no published recommended carnitine reference values. In the majority of cases, the estimated average daily carnitine requirements for an adult person amount to 20–200 mg, which is covered by diet and endogenous synthesis. Meat, fish and dairy products provide at least 80% of the required LC [87]. Based on rational dietary rules supporting good health, it is important to introduce carnitine-rich food [61,88] (Table 2), including carnitine supplementation. However, it should be taken into consideration that the bioavailability of LC from food is about four times higher than that from dietary supplements. Additionally, it should be taken into consideration that a high-fat, low-carbohydrate diet might be capable of boosting the endogenous synthesis of carnitine and its metabolites [89].

**Table 2 nutrients-12-01987-t002:** The amount of L-carnitine in product groups [61,88] modified.

Type of Food	Total L-Carnitine Content
**Ruminant meat**	**(mg/100 g)**
kangaroo meat	637
horseflesh	423
beef	98.2–139
beef steak	232
beef kidneys	31.0
beef liver	15.6
lamb	106–113
goat meat	95.0–99.0
pork	20.0–30.0
pork liver	10.7
**Poultry, bird meat**	**(mg/100 g)**
duck	73.0
pigeon	52.8
turkey	51.0
chicken	34.0
quail	29.1
pheasant	13.5
**Fish**	**(mg/100 g)**
salmon	5.96
zebrafish	2.80–8.95
yellow catfish	5.93
**Milk**	**(mg/100 mL)**
sheep	10.2–12.7
goat	4.50–7.50
cow	7.80–9.60
**Milk products**	**(mg/100 g)**
yoghurt	40.0
buttermilk	38.0
cottage cheese	22.5–26.6
sour cream	19.7
coffee cream	16.6
cheese	14.0–28.0
**Mushrooms**	**(mg/100 g)**
*Pleureotus ostreatus*—oyster mushrooms	53.0
champignon	29.8
*Cantharellus cibarius*—chanterelle	13.3
other mushrooms	1.00–6.00
**Vegetables**	**(mg/100 g)**
cucumber	4.45
cauliflower	3.26
carrot	3.73
maize	0.68
peas	0.60
**Fruits**	**(mg/100 g)**
avocado	1.72
guava	0.82
bananas	0.39
apples	0.29
orange	0.22

### 4.3. Supplementation with L-Carnitine and Its Derivatives 

In the prevention and treatment of patients with Alzheimer’s disease, supplementation with carnitine is essential for complementing intracellular and extracellular carnitine resources. In addition, LC supplementation is intended to facilitate the elimination of toxic metabolites that may interfere with mitochondrial homeostasis, thereby interfering with the production of cellular energy, further leading to increased ROS production in many neurodegenerative diseases. 

LC in the form of powder, fluid, tablets or capsules has been approved by the American Food and Drug Administration (FDA) for the treatment of primary and secondary carnitine deficiency. Experimental data obtained in in vitro and in vivo research did not demonstrate toxicity of LC. No side effects (including allergic reactions) were observed after the oral administration of LC in humans. However, some people using LC showed symptoms of alimentary tract intolerance (periodical nausea, diarrhea and tummy ache). People using large doses of LC may emit a fish body odor caused by the trimethylamine produced in the gut from LC by intestinal bacteria [56]. There were no published cases of LC intoxication. For the LC supplementation of the healthy adults, there were administered 250 mg to 2.0 g (highest safe dose) of LC daily, in several doses [90]. Daily LC doses greater than 2.0 g appeared to offer no advantage, since the gut mucosal absorption of carnitine appears to be saturated at about a 2.0 g daily dose [91]. A meta-analysis of 21 double-blind, randomized, placebo-controlled studies lasting from three months to one year showed that ALC either improved cognitive deficits or delayed the progression of cognitive decline. Improved cognitive function and delayed progression of cognitive decline were both statistically and clinically significant, with the magnitude of the effects increasing over time. Most studies used daily doses of LC of 1.5–2.0 g, which were well tolerated [92]. The treatment of ALC with doses of 2.25–3.0 g/day in patients with mild (initial) dementia caused by AD and vascular dementia (VD) led to a significant clinical improvement in patients with AD compared to in VD patients and placebo-treated patients [93]. In another study, 11 patients suffering from senile dementia of the Alzheimer’s type were treated intravenously with ALC at 30 mg/kg for 10 days, and 1.5 g/day per os for 50 days, in three daily doses. It was concluded that the intravenous and oral administration of multiple doses of ALC increases ALC plasma and CSF concentrations in patients suffering from AD, which suggests that ALC easily crosses the blood–brain barrier [94]. The bioavailability of LC in food supplements depends on the applied doses [59,89]. As a general guideline, the average therapeutic ALC dose is 1.0 g, given two to three times daily for a total of 2.0–3.0 g. No advantage appears to exist in giving an oral dose greater than 2.0 g of ALC at one time, since absorption studies indicate the saturation of GI receptors (receptors coupled with proteins G) at this dose [89]. The reported side effects of LC (especially at high doses) include agitation, headaches, diarrhea, nausea, vomiting, anorexia and abdominal discomfort, mostly of mild or moderate severity [95].

### 4.4. Choosing Proper Form of Carnitine Supplements

LC and its derivatives have been proposed as drugs or as adjuncts to conventional medicine for many conditions, including stable angina, intermittent claudication, diabetic neuropathy, kidney disease and dialysis, hyperthyroidism, male infertility, erectile dysfunction, chronic fatigue syndrome, AD and memory impairment [95]. Many specimens containing LC differing only in form (powder, liquid, tablets or capsules) are available on the market. Free or in combination with organic acids, e.g., citric, fumaric or orotic acids, LC given as a dietary supplement is perfectly bioavailable. A combination of LC and arginine facilitates the release of ammonia as an end product of protein and amino acid metabolism. It was reported that for epileptics, exceptionally beneficial is a combination of LC with taurine because taurine acts as a modulator of membrane excitability in the central nervous system by inhibiting the release of other neurotransmitters and decreasing the mitochondrial release of calcium [96]. When administering organic salts of LC, it should be taken into consideration that a given free carnitine content corresponds to a higher weight of drug [59,89]. However, healthy people should avoid the oral intake of LC at amounts higher than 1.0–2.0 g/day in 3–4 doses [97]. The bioavailability of LC from foods is 54–87% of the LC content and is dependent on the amount of LC in the meal. The absorption of LC from dietary supplements (0.5–6.0 g/day) is primarily passive, and the bioavailability is 14–18% of LC in the dose. LC that is unabsorbed in the gastrointestinal tract is mostly degraded by microorganisms in the large intestine [58]. 

#### 4.4.1. Pure L-Carnitine (LC)

A prevalent and the most economical form of L-carnitine supplementation is orally administered LC powder or solution. LC powder or solution is recommended for people taking care of their appearance and pursuing a reduction in or maintaining the proper level of their body weight. LC is also recommended for supporting the circulatory system by reducing the risk of ischemic heart disease and other circulatory disorders [98]. LC, as an organic osmoprotectant, has been proven to have protective roles against the production of proinflammatory mediators and apoptosis in primary human corneal epithelial cells exposed to hyperosmotic media, as well as in dry-eye patients [99,100]. It was reported that LC supplementation improved the depression state in patients undergoing hemodialysis [101]. A deficiency of carnitines in terminally ill HIV/AIDS patients requires supplementation. Significant reductions in serum lactate after LC supplementation may have clinical significance in patients taking certain antiretroviral drugs [102]. L-carnitine plays an important role in energy metabolism. Skeletal muscles store about 95% of the total of 20 g carnitine contained in the adult human body, but high-intensity physical exercise decreases the muscle’s carnitine content. It has been proven that L-carnitine supplementation may enhance athletic performance when coupled with physical exercise itself [103].

#### 4.4.2. Acetyl L-Carnitine (ALC)

L-carnitine acetylation increases L-carnitine’s hydrophobicity, which permits ALC’s crossing of the blood–brain barrier. ALC shows neuroprotective action on nervous cells, supports energetic metabolism and the regeneration of nerve cell structures, and alleviates mitochondrial dysfunction and apoptosis [104], improving memory and creativity. Several studies have suggested a beneficial effect of ALC on cognition and behavior in aging and AD subjects [23]. In AD patients, ALC improves clinical and cognitive functions in the short and medium term (3 and 6 months) in varied doses (1.5–3.0 g/day). Additionally, with 12 month treatment at a dose of 2.0 g/day, ALC slows down the deterioration of cognitive function in AD patients [19]. Other experimental data confirm the effectiveness of ALC supplementation in protecting against brain damage, e.g., the application of 100 mg/kg body weight of ALC reduced the volume of brain injury and improved victims’ behavior after traumatic brain injury. Additionally, ALC reduces oxidative stress and improves the function of mitochondrial membranes provoked by neurotoxic glutamate action [105]. The efficacy, safety and tolerability of ALC were studied during a double-blind, placebo-controlled, 12-week trial in patients with initial dementia caused by AD and vascular dementia (VD). The trial ended with the conclusion that ALC (carnicetine) can be recommended at doses of 2.25–3.0 g/day for the treatment of the early stages of AD and VD. ALC was well-tolerated [93]. Recently, the unique pharmacological properties of ALC have been confirmed, which allow us to look at this molecule as a representative of the next generation of antidepressants with a safe profile, especially for older people [106].

#### 4.4.3. Propionyl L-Carnitine (PLCAR)

Propionyl-L-carnitine (PLCAR), or L-carnitine esterified with propionic acid, is more stable and bioavailable than free carnitine, with significantly stronger action than free L-carnitine, especially in the circulatory system and cardiac muscle. PLCAR was recommended for the treatment of diseases of the peripheral arteries and other disturbances of the cardiovascular system [98,107]. According to some data, PLCAR increases the concentration of L-carnitine in muscles independently of insulin levels, which is beneficial during the application of low carbohydrate diets. It was proven that PLCAR prevents the peripheral neuropathy connected with diabetes or toxic chemotherapy, improving nervous conductivity and blood flow in peripheral nerves [81].

#### 4.4.4. Acetyl L-Carnitine Arginate (ALCA)

Damaged mitochondria are associated with decreased ATP production and increased reactive oxygen species production, both of which characterize AD patients. Arginine is a substrate for nitric oxide (NO) synthesis that improves the extension of blood vessel walls and improves blood circulation in the organism [108]. Acetyl L-Carnitine Arginate is an assistant to mitochondrial function, that helps to promote cell growth and cellular differentiation and may even play a role in the slowing the aging process. ALCA helps to boost mitochondrial energy production and promotes targeted benefits for the brain, heart and central nervous system. Supplementation with ALCA results in an increase in resting nitrate/nitrite levels in pre-diabetics, without any statistically significant changes in other substances connected with metabolic or oxidative stress (malondialdehyde, xanthine oxidase activity and hydrogen peroxide) [109]. Alpha-lipoic acid (ALA) (an eight-carbon saturated fatty acid)—a compound with strong antioxidative action, required by the pyruvate dehydrogenase complex for starting the tricarboxylic acids cycle—is frequently added to ALC. An important function of ALA in organisms is a redirection of anaerobic to aerobic metabolism in cells, preventing the acidification of the organism and enabling the generation of much more energy than can be generated by anaerobic metabolism [110]. It was reported that old rats that were fed with ALC and ALA significantly reduced their numbers of severely damaged mitochondria and increased the numbers of intact mitochondria in the hippocampus. The above results suggest that feeding ALC with ALA may also ameliorate age-associated mitochondrial ultrastructural decay in humans [110].

#### 4.4.5. Glycine-Propionyl- L-carnitine (GPLC)

Glycine improves the gut absorption and transfer through the intestinal walls of L-carnitine and facilitates the reaching of the circulation by L-carnitine. The combination of L-carnitine with glycine and propionic acid (GPLC) significantly improves the absorption and utilization of L-carnitine by cells. GPLC supports the production of nitric oxide (NO), an important substance facilitating blood circulation during physical training [111]. By stimulating NO synthesis, GPLC induced the distension of blood vessels, lowering pressure on the blood vessel walls. GPLC facilitates the faster and more efficient transport of fats to muscle cells (an important source of energy during prolonged physical exercise, when glycogen stores are exhausted) and facilitates the elimination of metabolic waste products. GPLC exhibits strong antioxidative properties that protect cells against the action of free radicals of oxygen and nitrogen. It was reported that GPLC significantly increases glutathione levels and decreases the levels of markers reflecting an increased speed of protein and lipid oxidation in humans [112]. GPLC increases the effectivity of citric acid cycle and inhibits lactate synthesis, prolonging the time of effective physical activity [113]. Some studies showed that GPLC significantly blocked D-galactosamine-induced pro-inflammatory cytokine (TNF-α and IL-6) production and, at the same time, inhibited the expression of α-smooth muscle actin, collagen-I and transforming growth factor-β. It has been demonstrated that GPLC has hepatoprotective effects against fulminant hepatic failure and chronic liver injury induced by D-galactosamine. Blommer et al. [114] determined the effect of GPLC on oxidative stress biomarkers at rest and on reactive hyperemia during the exercise of trained men. They found a decrease in lipid peroxidation with the oral intake of GPLC at rest, and in previously sedentary subjects. Whereas short-term ischemia–reperfusion in trained men results in a modest and transient increase in blood oxidative stress biomarkers, oral GPLC supplementation during short-term ischemia–reperfusion in trained men does not attenuate the increase in oxidative stress biomarkers.

#### 4.4.6. L-Carnitine-L-Tartate (LCLT)

L-carnitine-L-tartate (LCLT) contains L-tartate coupled with L-carnitine. L-tartate intensifies L-carnitine’s action by decreasing glucose absorption from the gastro-intestinal tract and decreasing the deposition of spare fats. LCLT is recommended for persons desiring to reduce fat tissue and improve efficiency and muscle force during training. Some research has shown that LCLT supplementation beneficially affects markers of hypoxic stress following resistance exercise. Muscle oxygenation was reduced by LCLT in the trial upper arm occlusion and following each set of resistance exercise. Despite reduced oxygenation, plasma malondialdehyde, a marker of membrane damage, was attenuated during the LCLT trial. The hypoxic stress was attenuated with LCLT supplementation [115]. The use of LCLT relieved the damage caused by metabolic stress and the hypoxic chain of events leading to muscle damage after exercise (reduced post-exercise serum levels of hypoxanthine, xanthine oxidase and myoglobin and perceived muscle soreness) [116]. In addition, a positive effect of LCLT on the endocrine system has been demonstrated. The supplementation of LCLT increased androgen receptor content in the muscle, which may result in increased testosterone uptake and thus enhanced luteinizing hormone secretion via feedback mechanisms, which may promote recovery post resistance exercise [117]. Chronic LCLT supplementation increased carbohydrate oxidation during exercise [118]. The influence of LCLT on markers of purine catabolism (hypoxanthine, xanthine oxidase and serum uric acid), circulating cytosolic proteins (myoglobin, fatty acid-binding protein and creatine kinase), free radical formation, and muscle tissue disruption after squat exercise was examined. Exercise-induced increases in plasma malondialdehyde (a lipid peroxidation product) returned to resting values sooner with LCLT supplementation than with a placebo. The above data indicate that LCLT supplementation is effective in assisting recovery from high-repetition squat exercise [119].

## 5. Conclusions

From the point of view of rational nutrition standards and the MIND diet favoring good health, it is beneficial to introduce nutritional products rich in L-carnitine and its derivatives or the supplementation of the diet with L-carnitine, and especially ALC, for the prevention and/or alleviation of dementia and other AD symptoms.

Despite the controversy concerning the consumption of red meat—based on the belief that a high consumption of red meat increases the risk of cancer, particularly colon cancer—a properly balanced diet should contain animal food. Excessive restrictions or the extreme avoidance of eating meat or dairy products in the human diet eliminates many bioactive substances necessary for the correct development and functioning of the organism. Therefore, it is necessary to implement diets with food products rich in carnitine and its derivatives.

It should be stressed that correct nutrition is an important element of lifestyle that may be an important factor for healthy, slow, favorable aging and delaying the development of neurodegenerative diseases including dementia and AD. We should keep in mind that a correct diet rich in vegetables with low starch content, fruit with low glycemic indices, cereals, legumes, vegetable oils and sea fish with reasonable amounts of meat and dairy products is essential for our health and lowers the risk of dementia and AD.

## Figures and Tables

**Figure 1 nutrients-12-01987-f001:**
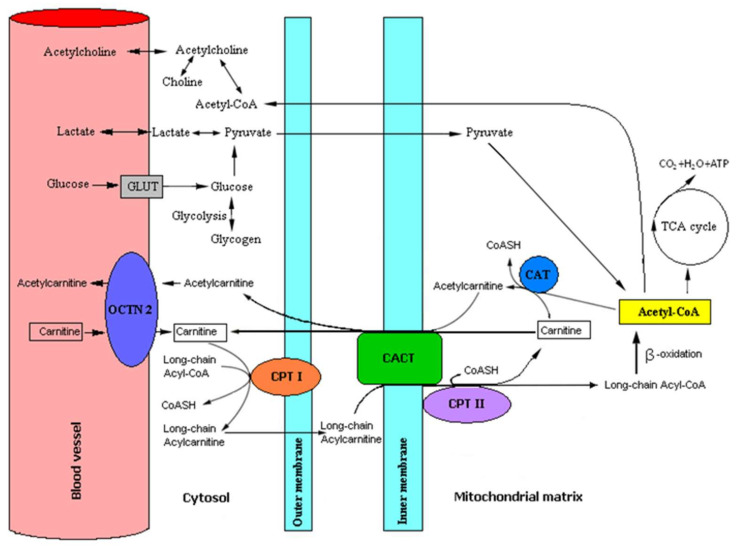
The role of carnitine and other substances in the brain’s energy supply. Abbreviations: GLUT, glucose transporters; OCTN2, sodium-dependent carnitine organic cation transporter; CPT I, carnitine palmitoyltransferase I; CPT II, carnitine palmitoyltransferase II; CACT, carnitine acylcarnitine translocase; CAT, carnitine acetyltransferase; CoASH, coenzyme A; Acetyl-CoA, acetyl coenzyme A; TCA cycle, tricarboxylic acid cycle.

**Table 1 nutrients-12-01987-t001:** Type and frequency of consumption of foods in the Mediterranean-DASH Intervention for Neurodegenerative Delay (MIND) diet pattern with a role in Alzheimer’s prevention [26,48], modified.

No.	Type of Food	Components	Intake Categories	Characteristics
1.	green, leafy vegetables	kale, spinach, kohlrabi, different varieties of lettuce, cooked greens and salads	≥6 servings/week	vegetables rich in vitamins C and A
2.	all other vegetables	celery, cabbage, beets, cucumbers, cauliflower, zucchini, tomatoes, leeks, garlic and onion	1 serving/day	to choose non-starchy vegetables with a lot of nutrients and a low number of calories
3.	berries	strawberries, blueberries, raspberries, blackberries	≥2 servings/week	a source of antioxidants
4.	nuts and almonds	pineapple, pistachios, macadamia, pecans, peanuts and Brazilian walnut	≥5 servings/week	are a source of unsaturated fatty acids and antioxidants; contain vitamins E, B_1_ and PP; and reduce the level of “bad” cholesterol
5.	whole grains	oatmeal, quinoa, brown rice, whole-wheat pasta and 100% whole-wheat bread	≥3 servings/day	a source of fiber, folic acid, vitamin B_3_, iron, zinc, magnesium and phosphorus
6.	fish	salmon, sardines, trout, tuna and mackerel	≥1 serving/week	high amounts of omega-3 fatty acids
7.	poultry	chicken or turkey	≥2 servings/week	fried chicken is not encouraged on the MIND diet
8.	beans	lentils, soybeans, string beans, broad beans, green peas, chickpeas and white beans	≥4 servings/week	a source of fiber, protein, vitamins and minerals
9.	olive oil	cold pressed oils		a source of vitamins A, D, E, and K; polyunsaturated fatty acids; use fat for long frying at a high smoke point
10.	milk, dairy products	low-fat: milk, cheese, yoghurt, buttermilk, kefir and cottage cheese	≥2 glasses/day; 280–400 g semi-skimmed cheese or 1 slice (30 g) of yellow cheese.	a source of protein and minerals: calcium, potassium, phosphorus, magnesium, zinc, manganese, iron; high in vitamins: B_2_, B_12_, A, D, E, and K and probiotics
11.	wine	red and white	≤1 glass serving/day	both red and white wine may benefit the brain; red wine is recommended because a lot of research has focused on the red wine compound resveratrol
	beer	non-alcoholic beer	regular beer consumption is not recommended for some risk-group populations (pregnant, children, people affected by liver diseases)	source: carbohydrates, protein/amino acids (proline, glutamic and aspartic acid, glycine, alanine), minerals (fluoride, potassium, phosphorus, calcium, sodium, magnesium, silicon), vitamins (B1–B6, folic acid), and other compounds, such as polyphenols

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
