# Peer review of "Preventive Role of L-Carnitine and Balanced Diet in Alzheimer’s Disease"

_nutrients, 2020, doi:10.3390/nu12071987_

Round 1

Reviewer 1 Report

L-Carnitine, a naturally occurring compound found in most mammalian tissues including brain, functions to transport activated long chain fatty acids into the mithocondria, for degradation by beta oxidation.  Interest in the therapeutic potential of L-carnitine and acetyl-L-carnitine  for neuroprotection is increased in the last few years.

The paper submitted as is, presented missing tables and figures (mentioned in the ms). In addition the paper needs an extensive revision of the language and the style because is not well presented/nor organized

Finally subparagraph 4.4.1 is touching not only AD but also other disorders, mentioning several combinations of supplements without any link to AD, such as for example Spirulina and L-Carnitine or Taurine and L-Carnitine and others with no relation to the object fo the review. 

Author Response

Thank you for valuable comments and suggestions.

 Thank you for valuable comments and suggestions.

  1. The paper submitted as is, presented missing tables and figures (mentioned in the ms). Answer: To the attached manuscript we have attached the missing two tables and one figure.
  1. In addition the paper needs an extensive revision of the language and the style because is not well presented/nor organized. Answer: The text was checked and corrected  by the native speaker (marked in yellow).
  1. Finally subparagraph 4.4.1 is touching not only AD but also other disorders, mentioning several combinations of supplements without any link to AD, such as for example Spirulina and L-Carnitine or Taurine and L-Carnitine and others with no relation to the object to the review.  Answer: According to your suggestion, in the paragraph 4.4.1 we removed supplements that have not immediate  connection with AD such as Spirulina L-Carnitine, Taurine-L-carnitine  and L-carnitine fumarate. 

Reviewer 2 Report

Recommendation: Minor revisions

This manuscript by Alina and coauthors summarized the literature investigating the prevention or alleviation of AD, by introducing an appropriate carnitine-rich diet, dietary carnitine supplements and MIND diet.  Introcution food stuffs rich in carnitine and MIND diet or carnitine supplementation of the AD patients may improve theirs functioning in every day life.  Especially detailed the physiological properties of L-carnitine and the forms of carnitine supplements.  The author have made organized, logical and legible summary and such an comprehensive, detailed review is necessary for studying the preventive role of L-carnitine and balanced diet in Alzheimer disease. In view of this, the manuscript needs a minor revise.

Some questions and suggestions:

  1. Some schemes or Figures could be used in the paper to better illustrate.
  2. The conclusion part is too short. Some more convincing and suggestions need to be added to the conclusion.
  3. Lines 600 and 601, the format is not consistent with other references.
  4. Lines 610, 627, 649, 660, 664, 839, and 883, list all the authors.

Author Response

Thank you for valuable comments and suggestions.

  1. According to the Reviever suggestion, schemes or Figures could be used in the paper to better illustrate.

Answer: According to the Reviever suggestion to the final text we added 2 tables and one figure.

  1. The conclusion part is too short. Some more convincing and suggestions need to be added to the conclusion.

Answer: Final conclusions were expanded. There were added  some sentences and suggestions concerning healthy  MIND diet, and considered meat consumption as valuable origin of carnitine in the body. 

  1. Lines 600 and 601, the format is not consistent with other references.

Answer: According to the  editorial rules  for documents coauthored by a large number of persons (more than 10 authors), we  cited the first ten authors, and we  added ‘et al.’

For article numbered ‘1’ containing 9 pages in the PDF version, we added 9 pages (bold) and doi number: Strafella, C.; Caputo, V.; Galota, M.R.; Zampatti, S.; Marella, G.; Mauriello, S.; Cascella, R.; Giardina, E. Application of precision medicine in neurodegenerative diseases. Front. Neurol. 2018, 9,  701:1-701:9, doi: 10.3389/fneur.2018.00701.

  1. Line 610

Answer: Currently line 592. We added doi number (bold): Schliebs, R; Arendt, T. The cholinergic system in aging and neuronal degeneration. Behav. Brain Res. 2011, 221, 555-563. doi: 10.1016/j.bbr.2010.11.058.

  1. Line 627

Answer: Currently line 613. We listed all 5 authors and added doi number:  Szutowicz, A.; Bielarczyk, H.; Jankowska-Kulawy, A.; Pawełczyk, T.; Ronowska, A. Acetyl-CoA the key factor for survival or death of cholinergic neurons in course of neurodegenerative diseases. Neurochem. Res. 2013, 38, 1523-1542, doi: 10.1007/s11064-013-1060-x.

  1. Line 649

Answer: Currently line 640. We  changed the order of first and last names (bold):  Zhou, P.; Chen, Z.; Zhao, N.; Liu, D.; Guo, Z.Y.;, Tan, L.; Hu, J.; Wang, Q,.; Wang, J.Z.; Zhu, L.Q. Acetyl-L-carnitine attenuates homocysteine-induced Alzheimer-like histopathological and behavioral abnormalities. Rejuvenation Res. 2011, 14(6), 669-679, doi: 10.1089/rej.2011.1195

  1. Line 660

Answer: Currently line 656. We added information concerning number of pages (bold): Sánchez-Muniz, F.J.; Macho-González, A.; Garcimartín, A.; Santos-López, J.A.; Benedí, J.; Bastida, S.; González-Muñoz, M.J. The nutritional components of beer and its relationship with neurodegeneration and Alzheimer's disease. Nutrients 2019, 11, 1558:1-1558:36, doi: 10.3390/nu11071558.

  1. Line 664

Answer: Currently line 660. We added information concerning number of pages (bold): de Boer, B.; Hamers, J.P.H.; Zwakhalen, S.M.G.; Tan, F.E.S.; Verbeek, H. Quality of care and quality of life of people with dementia living at green care farms: a cross-sectional study. BMC Geriatr. 2017, 17(1), 155: 155:1-155:10, doi: 10.1186/s12877-017-0550-0.

  1. Line 839

Answer: Currently line 870. We added information concerning number of pages (bold): Chiechio, S.; Canonico, P.L.; Grilli, M. L-Acetylcarnitine: a mechanistically distinctive and potentially rapid-acting antidepressant drug. Int. J. Mol. Sci. 2017, 19(1), 11: 11:1-11:13, doi: 10.3390/ijms19010011.  

  1. Line 883

Answer: The 4.4.9  chapter and position 127 in the list of the literature were removed.

Finally, all literature references were  reviewed and missing data were attached.